# High-Speed Manipulation of Microobjects Using an Automated Two-Fingered Microhand for 3D Microassembly

**DOI:** 10.3390/mi11050534

**Published:** 2020-05-24

**Authors:** Eunhye Kim, Masaru Kojima, Yasushi Mae, Tatsuo Arai

**Affiliations:** 1Department of Systems Innovation, Graduate School of Engineering Science, Osaka University, Osaka 560-8531, Japan; 2Faculty of Engineering Science, Kansai University, Osaka 564-8680, Japan; mae@kansai-u.ac.jp; 3Global Alliance Laboratory, The University of Electro-Communications, Tokyo 182-8585, Japan; tarai118@jcom.zaq.ne.jp; 4Beijing Advanced Innovation Center for Intelligent Robot and Systems, Beijing Institute of Technology, Beijing 100081, China

**Keywords:** 3D assembly, automatic releasing, local stream, high-speed motion, micromanipulation, tissue engineering

## Abstract

To assemble microobjects including biological cells quickly and precisely, a fully automated pick-and-place operation is applied. In micromanipulation in liquid, the challenges include strong adhesion forces and high dynamic viscosity. To solve these problems, a reliable manipulation system and special releasing techniques are indispensable. A microhand having dexterous motion is utilized to grasp an object stably, and an automated stage transports the object quickly. To detach the object adhered to one of the end effectors, two releasing methods—local stream and a dynamic releasing—are utilized. A system using vision-based techniques for the recognition of two fingertips and an object, as well automated releasing methods, can increase the manipulation speed to faster than 800 ms/sphere with a 100% success rate (*N* = 100). To extend this manipulation technique, 2D and 3D assembly that manipulates several objects is attained by compensating the positional error. Finally, we succeed in assembling 80–120 µm of microbeads and spheroids integrated by NIH3T3 cells.

## 1. Introduction

Recently, tissue and organ transplantation has become very important work. In spite of advances in medical technology and an increased awareness of organ donation and transplantation, the need continues to grow. To overcome these problems of transplantation, in recent decades, tissue engineering—the development of complex tissues and organs—has grown as a new scientific field. For example, bioprinting [1,2,3], cell sheet engineering [4,5,6], and automated robotics approaches on the microscale [7,8,9,10] have been researched to build 2D and 3D structures. Among these methods, the manipulation of microobjects using a robotics system is an appropriate solution for building complex tissues or organs because it can assemble complex structure composed of different materials.

Manipulation-applying robotic approaches have been discussed. It can be divided into two categories: noncontact and contact manipulation. Noncontact manipulation utilizes field forces such as acoustic wave [11,12,13], dielectrophoresis [14], optical tweezers [15], and local flow [16,17]. Contactless manipulation is able to handle multiple objects at the same time, and is not affected by adhesion forces. However, it cannot perform physical operation tasks and struggles to assemble objects as a 3D structure. On the other hand, contact manipulation directly contacts target objects by employing several actuation, for example, a vacuum tool [18], voltage control [19], piezoelectric actuator [7,20], and plunging mechanism [21]. Although this method can be applicable for building 3D microsystems and devices [22], the main difficulties arise in the releasing of microobjects due to strong adhesion forces.

In this research, we apply the pick-and-place method that manipulates objects one by one, which makes the assembly time proportional to the number of objects. To reduce the assembly time, high-speed manipulation using one object should be conducted, allowing the manipulation to be extended to an assembly system that manipulates many objects. In addition, this manipulation should be conducted in a liquid environment for controlling biological cells. The target of this research is 100 µm spheroids composed by NIH3T3 cells.

To achieve high-speed manipulation in liquid using the pick-and-place method, there are two main problems. First, owing to the strong adhesion forces in a microscale environment, the release and accurate placing of microobjects is too difficult. Second, in a liquid environment, a drag force can be generated by the movement of the manipulator because of the high viscosity of the liquid. In particular, the faster the movement, the higher the drag force. Thus, it is necessary to reduce this influence of flow for high-speed manipulation and precise positioning. To solve these two difficulties—strong adhesion force and drag force—special methods for high-positioning and high-success rates are required.

Based on environmental conditions, the adhesion forces and drag forces are different. In air, the adhesion forces are stronger than in water, while the drag forces are ignorable. On the other hand, in liquid, the adhesion forces are weaker than in air, whereas the drag forces are strong enough to hinder precise manipulation. However, the adhesion forces are still strong, so the microobject will adhere to the end effector during the manipulation process [23,24,25,26]. Thus, in a liquid environment, both forces should be considered. The forces cause several difficulties. First, the adhesion forces impede the release and accurate positioning of a target object. Second, drag forces can reduce the speed and the success rate of transportation. Third, drag forces make it difficult to arrange objects at close distances for constructing 2D and 3D structures.

Recently, many automated manipulation approaches were studied to manipulate objects at high speed. In the air environment, Xie et al. manipulated 3−4 µm spheres at high speed (48 s/sphere) by transporting at 9 µm/sphere. To pick and place the objects, a nanotip gripper with multifeedback was applied. The researchers built 3D pyramids using four objects successfully [27]. Zhang et al. transported 7.5−10.9 µm spheres at high speed (82 µm/s) to 77 µm/sphere and their manipulation speed was 6 s/sphere. A microgripper with a plunger was utilized for active release. The researchers constructed 2D patterns such as alphabets and circle with high position accuracy [21]. From previous works, strong adhesion forces and low drag forces makes it easy to transport objects without loss of the object in spite of the high-speed transportation. In addition, 2D and 3D assembly have been accomplished by manipulating several microobjects regardless of the flow effect.

On the other hand, in liquid, high drag forces restrict the transporting speed and increase the loss of object retention. Moreover, they make it difficult to arrange several objects as 2D or 3D structures. For example, Lu et al. deposited biological cells in 15 s using partial-cell aspiration. The transporting speed was approximately 0.2 mm/s when transferring 1 mm. Although their speed was faster than in previous works [21,27], they were unable to achieve the maximum speed (1 mm/s) owing to the loss of the object during high-speed transportation. In addition, they deposited cells in an array of microwells to position the cell in 2D arrays. In this case, a complicated fabrication process is required to create microwells [28].

Another example is the 2D manipulation of 55 µm microbeads. The researchers achieved high-speed manipulation in 1 s with an 84% success rate. The transportation speed was 2.3 mm/s with a 95% success rate. However, the transporting distance was too short (60 µm), and the target movement was limited in 2D. To release the transferred objects, the researchers generated vibration by applying high frequency. This method could release successfully, but accurate positioning after the release was not achieved [20]. Table 1 shows a comparison of the high-speed manipulation in air and liquid.

From previous works, we understand that a stable transporting system and special releasing method are required in order to achieve high-speed manipulation that is faster than in previous works. In particular, to build a 3D structure using multiple microobjects at high speed, stable grasping, a high speed transporting system that transfer objects them without loss of the objects by covering a large workspace, and dexterous motion for releasing different sizes of objects on the desired position are necessary.

In our previous works, we proposed a two-fingered microhand for manipulation including grasping, transporting, and releasing [20,29,30]. A parallel mechanism was utilized to make dexterous motion for grasping and releasing of objects. The right end effector using a parallel mechanism generates the motion, while the left end effector is not actuated. For this, we propose two releasing strategies regarding the two cases where the object is adhered to the left and right end effectors [30,31,32]. However, the early works have several limitations to assemble 3D structure at high speed.

In this paper, we significantly improve the early version of the studies. The four main enhancements of this paper are the following: (1) improvement of a transporting system by changing the system; (2) increase in success rate and a position accuracy of the releasing task by analyzing local stream; (3) application of a vision based automatic manipulation; (4) assembly of 2D and 3D structure at a high speed that is faster than previous works; (5) manipulation of NIH3T3 spheroids for tissue engineering.

## 2. Materials and Methods

### 2.1. System for Micromanipulation

In this system, a two-fingered microhand is utilized for grasping, transporting, and releasing objects. The overall system of the microhand is displayed in Figure 1a. In order to not only grasp different sizes of microobjects but also transport the grasped objects for long distances, we divide the motion of the microhand to two parts: a global motion and a local motion. The global motion handles the movement of the substrate for transporting targets. In contrast, the local motion manipulates the right end effector controlled by a compact parallel link for grasping and releasing variously sized objects.

In our system, the right end effector generates 3D active motion, while the left end effector does not move but holds the object for transporting. One adjustment stage manually adjusts the right end effector, which can move 6 mm in a 3D direction with a resolution of 3 µm. To manipulate microobjects with different diameters, a large distance between the two end effectors should be realized. The proposed microhand is able to manipulate objects from 10 µm to 120 µm of diameter. Two end effectors of the microhand and the target objects are observed under an inverted optical microscope (IX71, Olympus, Tokyo, Japan) using an objective lens. We sharpened glass needles by heating and vertically pulling a glass rod (NARISHIGE G-1000, Narishige, Tokyo, Japan) of 90 mm in length and 1 mm in diameter. The end effectors are approximately 23 mm in length, 1 mm in diameter, and less than 1 µm in diameter at the end effector tip.

Figure 1b shows the architecture of the micromanipulation system. We set up an automated manipulation system by using two operating systems: a Windows PC (Intel Core i7 CPU at 2.93 GHz with 4 GB of RAM, Intel, Santa Clara, CA, USA) and a Linux PC (Dell, XPS600, Pentium 4 at 3.80 GHz, Dell, Landrock, TX, USA). The Windows PC manages the overall system for high-speed manipulation. It controls a motorized Figure 1b shows the architecture of the micromanipulation system. We set up an automated manipulation system using two operating systems: a Windows PC (Intel Core i7 CPU at 2.93 GHz with 4 GB of RAM) and a Linux PC (Dell, XPS600, Pentium 4 at 3.80 GHz). The Windows PC manages the overall system for high-speed manipulation. It controls a motorized stage that moves a substrate for the global motion and a piezo actuator attached to an objective lens for autofocusing, and provides images of the two end effectors and targets captured by a high-speed camera (Photron FASTCAM MC2, Photron, Tokyo, Japan). In addition, it communicates with the Linux PC, which manipulates the parallel link for local motion through a TCP/IP connection. The motorized stage, actuated by three stepping motors (SGSP-13ACT-B0, Sigmakoki, Tokyo, Japan), is controlled by three motor drivers (SG-55MA, Sigmakoki, Tokyo, Japan) via an I/O board. The stage can be moved up to 13 mm on the XYZ-axis at high speed (1 mm/s). In addition, the position of the piezo actuator determines the height of the objective lens that can focus the images of the objects and two end effectors via a D/A board (Contec DA12-16(PCI), CONTEC, Osaka, Japan) and a piezo controller (ENV 150, ENT 150/20, Piezosystem Jena, Tokyo, Japan). The Linux PC controls the right end effector’s motions via a parallel link actuated by three piezoelectric (PZT) actuators (NEC TOKIN, AE0203D16, Shiroishi, Japan). The 3D position of the right end effector is determined by the displacement of the PZT actuators through a D/A board (Contec DA16-16(LPCI)L) and a piezo amplifier (MATSUSADA, HJPZ-0.15Px3, Tokyo, Japan). Strain gauges measure the displacement of the PZT actuators via an amplifier (Kyowa MCD-16A, Kyowa, Tokyo, Japan) and A/D converter (Contec AD16-16(PCI)EV).

### 2.2. Accurate and High-Speed Manipulation

Figure 2 describes the manipulation process of an object. First, a target object and finger-tip are detected by image processing. Using these positions of the target and fingertip, the moving distance for grasping is calculated. Subsequently, the target approaches the end effectors by moving the motorized stage. After the object is grasped, the automatic stage is moved in order to transport the target to a desired location (60 µm away) at 1 mm/s. At that time, the microhand picks up the target by lowering the stage to 20 µm to fix the releasing height. To compensate the error during the grasping and transportation of the target, the center of the grasped object is detected and then the stage is moved based on the calculated error. Finally, to release the target object, the parallel link opens the right end effector. After moving the right end effector, the target can be released or adhered to one of the end effectors. If the object is adhered to one of the end effectors, we apply high-speed motion by a parallel mechanism.

#### 2.2.1. Object and Fingertip Detection for Stable Grasping

To grasp a target object firmly, it is important to detect the 3D position of the target and end effectors. First, we can grasp the middle parts of objects by detecting the Z-position of the target object and the fingertip. The target object and the left fingertip are detected using a high-speed camera and a piezo actuator. The high-speed camera can capture the image at 2000 frames per second. The piezo actuator moves the objective lens over a displacement of 90 µm in the Z-direction for a top-down scan of the target object and the end effectors. To search for the optimal position of the target in the Z-direction, the piezo actuator moves the objective lens by 1 µm in the Z-direction. The high-speed camera captures the image whenever the lens moves. Thus, 90 images are successively captured and regularly spaced focal planes are acquired. By comparing stacked images, the best focused image can be found. The best focused image of the object and the left end effector are found using the depth from border intensity variation (DFBIV) algorithm and template matching, respectively. A detailed explanation of the algorithm for detecting the 3D position of the fingertip and Z-position of the target object is in [20].

The 2D position of the target object is recognized as follows. First, the adaptive threshold is applied to minimize the effect of lighting. After that, contours are formed, and bounding rectangles are drawn based on the formed contours. If the ratio of the contour height and width is between 0.8−1.1 and the object size is from half of the object to three times the object, the object is considered to be the target. On the other hand, if the ratio is higher than 1.1 and lower than 2, the object is analyzed by a Hough transform to find the circles.

Using the detected position of the center of the target object ($Obj_{x}$,$\;Obj_{y}$) and the fingertip ($f_{x}$,$\;f_{y}$) displayed in Figure 3, we can calculate the moving distance of the object (Δx, Δy) for grasping. Then, the object approaches the space between the two end effectors by moving the motorized stage. To grasp the object, the parallel mechanism closes the right end effector. The grasping distance is determined by the size of the target object.

#### 2.2.2. High-Speed Transportation in Liquid

Compared with the environment in air, the high dynamic viscosity and the weak adhesion force make high-speed transportation in liquid difficult. In the environment of liquid, the influence of the fluid flow is larger than in air owing to the high dynamic viscosity. Based on Stokes’ drag Equation (1), the influence of the flow can be estimated [23]. $F_{d}$ in Equation (1) indicates the hydrodynamic force of the fluid on a sphere. *μ* represents the dynamic viscosity of the medium (air: $1.85\times 10^{-5}$, water: $1\times 10^{-3}$ (Pa·s)), $R_{b}$ is the radius of the object, and *V* is the relative velocity of the fluid with respect to the object.
(1)$$F_{d}=-6\pi \mu R_{b}V$$

From Equation (1), we understand that high dynamic viscosity is a reason for a strong drag force, which makes it difficult to control the microobjects precisely and quickly. To attain high-speed transportation of microobjects without loss of the object in liquid, it is required to hold the target object firmly.

In air, the grasped object is not easily detached from the end effector owing to the strong adhesion force, which allows for ease of transport. For example, manipulation methods in air succeeded in transporting objects at high speed without loss of the transferred object [21,27,33]. However, in liquid, transportation of a target object at high speed is more difficult than in air because the drag force exerted on the transferred object increases as the velocity of the motion increases. The drag force also causes the loss of the transferred object. Thus, previous manipulation methods in liquid chose a slow speed for the transporting of objects.

For example, optical tweezers can manipulate cells precisely, but rapid cell transportation is too difficult. The cells can easily escape from the trap since the trapping force is weaker than the drag force during high-speed movement. Thus, the maximum speed is approximately 10 µm/s [15,34,35]. Lu et al. transported cells at slower speed (0.2 mm/s) than the maximum speed of the stage (1 mm/s) because the cells partially aspirated to the pipette were frequently detached from the pipette at the maximum speed [36]. Thus, we understand that stable grasping is a significant factor in high-speed transportation. In our case, we utilized a two-fingered microhand with dexterous motions and grasped objects stably by detecting the 3D position of the target.

To transfer the target object, there are two methods: moving two end effectors and moving the substrate (Figure 4). Similar to the high speed end effector’s motion, the residual vibration can be increased during high-speed transportation, causing the loss of the transferred objects. For the first case, moving end effectors, as shown in Figure 4a, controls the absolute position and makes the positioning intuitive. However, this method can generate vibration at the two end effectors while transporting objects, and the transporting distance is limited to the visible space. For the second case—moving the substrate, as described in Figure 4b, the vibration is considerably reduced when transporting objects, although the position is determined based on other objects. In addition, the system can move long distances over the visible space.

Many previous works moved the end effectors for transportation. In air, these studies transferred microobjects without the loss of the object despite the vibration [21,27,33]. However, in liquid, the residual vibration (more than 5 µm) leads to loss of the objects, although the researchers applied a control algorithm to suppress the vibration [20]. This is because many studies in liquid transported the objects by moving the substrate [15,34,35,36]. In our system, we moved the substrate for the high-speed transportation of objects, and then reduced the vibration from more than 5 µm to less than 0.5 µm at 1 mm/s. Thus, we transported various sizes of objects (40−120 µm) without loss of the grasped objects.

#### 2.2.3. Compensating Error during Grasping and Transporting

To compensate the positioning error during the grasping and transporting of an object, an additional method is required. Before releasing the grasped object, the error is calculated. For a high positioning accuracy of manipulation, the center of the grasped object is detected. First, we try to detect the object using the existing method described in the previous subsection. However, it is difficult to recognize the central position of the object using a bounded rectangle because the grasped object and two end effectors are detected together.

To detect the object only, we remove the image of the two end effectors based on the finger position because the left finger position is not changed from the initial position in our system. Figure 5a shows a binary image of the grasped object with fingers and the result of the detected objects. After removing the finger image, we can show the grasped object without fingers, as in Figure 5b. Finally, the object is detected correctly. By comparing the object position and the desired position, we can calculate the positional error and then move the stage based on the calculated distance.

#### 2.2.4. Release and Precise Positioning of Objects

To release an object successfully and precisely, an automated system is applied. In our releasing approach, the release strategy depends on the state of the object. To analyze the state of the target object, the position of the left fingertip and the center of the target object are detected (Figure 3). By calculating the distance between the tip of the left end effector and the center of the target, the releasing strategy can be determined. We calculate the distance in the X- and Y-direction between the tip of the left end effector and the center of the target object ($Obj_{x}-f_{x}$ and $Obj_{y}-f_{y}$ in Figure 3). Using the distance in the X-direction, the state of the object after the release can be analyzed. If the distance in the X-direction is longer than the sum of the radius of the target object and moving distance of the right end effector for release without any special method, the object is adhered to the right end effector. On the other hand, if the distance in the X-direction is shorter than the radius of the object, the object is adhered to the left end effector. The other case is that the object is released without any special method. To analyze whether the object is effectively transported, we calculate the distance in the Y-direction. If the distance is longer than 0 and shorter than the diameter of the target, the object is located in space between the two end effectors.

After the release without any special method, the state of an object can be divided into three cases. First, the object is released. Second, the object is adhered to the left end effector. Third, the object is adhered to the right end effector. For the first case in which the object is released without any special method, we can skip this process and apply high-speed motion, to minimize the processing time. On the other hand, when the object is adhered to the left end effector, the releasing method using a local stream can be utilized, as shown in Figure 6a. To place the object at the desired position precisely, a counterclockwise circular motion with an amplitude of 6 µm and frequency of 100 Hz is employed [31,32]. When the object is adhered to the right end effector, dynamic releasing is used to detach the target, as shown in Figure 6b. In this case, a clockwise circular motion with an amplitude of 4.8 µm and frequency of 100 Hz is applied for precise positioning [30,32]. The performance time of both circular motions is 50 ms.

However, the target sometimes sticks to the left end effector, although the end effector exerts a local stream using high-speed motion. To detach the target from the left end effector every time, the right end effector generates a local stream again. At this time, a stronger local flow than before is necessary to release the target. Using experimental results and vortex theory, we formulated simple equations to analyze the flow velocity [37]. The amplitude can be estimated by calculating the velocity. The velocity can be calculated by Equations (2) and (3). $C_{1}$, $C_{2}$, and $C_{3}$ indicate the rate constant, and *r* is the radius of the circular motion. $C_{1}$, $C_{2}$, and $C_{3}$ are 0.0203, 0.0355, and −0.034 respectively. f represents the end effector frequency, and *A* is the amplitude.
(2)$$\omega =C_{1}\frac{f^{2}A^{2}}{r^{2}}$$
(3)$$v_{y}=\frac{C_{2}f^{2.2}A+C_{3}f^{2.7}}{r}$$

The force can be estimated by Equation (1). If the distance between the object and the end effector that generates the releasing motion is 30 µm when the end effector motion has an amplitude of 6 µm and a frequency of 100 Hz, the flow velocity is 0.266 mm/s. Since the target objects are 100 µm microbeads and the dynamic viscosity of water is $1\times 10^{-3}$ (Pa·s), the force is $5.01\times 10^{-10}$ N.

To release an adhered object that is not detached in spite of the local stream, a stronger force is applied by changing the amplitude of the high-speed motion. To release the target from the left end effector without fail, we tried to increase the amplitude of the high-speed motion by 1 µm. As a result, a 9 µm amplitude of the end effector motion is selected. The flow velocity is 0.548 mm/s, and the force is $1.03\times 10^{-9}$ N. Finally, the objects are released every time. Figure 7a shows the releasing process in order to achieve a 100% success rate of releasing objects. To minimize the releasing time, the object state is only analyzed when a local stream is exerted. Thus, according to the adhered state, the processing time is different.

Figure 7b shows the manipulation process when a second local flow is applied because the object is still attached to the left end effector after the first local stream. The right end effector generates the local stream by the high-speed motion. However, the object is still adhered to the left end effector. To separate the adhered object, the right finger again generates a local stream using a larger amplitude of the end effector motion. Finally, the object is released from the left end effector and then moves to the desired location.

#### 2.2.5. Manipulation Process for Assembling Objects

To cover a large workspace for several objects, the assembly process is as follows: Figure 8 displays the assembly process for manipulating two objects. Our goal is to arrange two objects as a 1D string. Figure 8a shows one placed object and two objects to be manipulated. The first object, indicated as “1”, is already manipulated to the desired position, while the second object (“2”) and the third object (“3”) are target objects to be manipulated. The white circle indicates the desired position of the second object. The rectangular with red border indicates the visible space by a microscope. In this case, the second object is not located in the visible space.

To find the second object that was not visible the first time, the microhand moves to the left (−X-direction) by changing the stage movement. After moving the stage, we can find the second and the third objects. In our system, the object placed on the right side is operated first. In Figure 8b, the hand grasps object “2” because the object is located on the right side. After gripping the target, the microhand transports the target to the desired position by moving the motorized stage (Figure 8c). Finally, the target is released by a parallel link, and then the two objects are arranged as a 1D string (Figure 8d). To manipulate the third object, we move the stage to the left again and repeat the process. By using this approach, we can manipulate many objects outside of the visible space determined by the microscope and cover a larger workspace than previous works regarding high-speed manipulation [20,21,27,28].

## 3. Results and Discussion

### 3.1. Automated Manipulation of an Object

To analyze the placing accuracy of different sizes of objects after the release, experiments are conducted using 55 µm and 100 µm microbeads. The manipulation process is same as the Figure 2 and the experiments are repeated 20 times. In our system, the final position of the object is different from the releasing motion. Thus, we first inspect the object state and apply motions, and then analyze the placing position of objects according to the releasing motion.

For manipulating 55 µm microbeads, we use 20× objective lens and make a visual space as 256 µm × 256 µm. As a result, the objects are adhered to the left end effector ten times, adhered to the right end effector five times, and released without any special method five times. After applying the local stream, the objects are still adhered to the left end effector three times. Figure 9a shows the result of the placing position of 55 µm microbeads. From the experimental results, the placing position can be analyzed. All objects are located in the right side of the desired position. Notably, the objects released by the local stream are moved to the root of the end effector. This is because the objects detached by the circular motion are not only rotated around the end effector but also move to the root of the end effector.

To manipulate 100 µm microbeads, we use a 10× objective lens and create a visual space of 512 µm × 512 µm. As a result, the microbeads are released without any special method ten times, objects adhere to one of the end effectors ten times. To detach the object, dynamic releasing is applied twice, and a local stream is exerted eight times. After applying a local stream, the object is still adhered to the left end effector for one time. The placement of the 100 µm microbeads is shown in Figure 9a. The objects detached by dynamic releasing are located to the right side of the desired position. In addition, the objects (except for the object released by dynamic releasing) are moved toward the Y-direction and the right side of the desired position. From the experimental results, we can compensate the positional change during release according to the motion. To extend the manipulation of an object to the microassembly, it is necessary to compensate the positional change of the objects after the release.

Table 2 lists the average required times for manipulation of two objects. Both objects are manipulated successfully without any failure cases. In addition, the manipulation time is less than 800 ms including transporting the object by 60 µm in the X-direction. The average time of the grasping tasks are 225 ms for 55 µm microbeads and 270 ms for 100 µm microbeads, in addition, the transporting time is 105 ms when the stage moved 60 µm. After transporting the target object, the releasing task is performed according to the motion. The average time of the releasing tasks of 55 µm and 100 µm microbeads is 382.5 ms and 330 ms, respectively. Thus, our approach is faster than that of the previous work [20], which was the highest speed studied in the literature. Figure 10 show time-lapse images of the automated manipulation of an object.

### 3.2. Microassembly of 2D and 3D Objects

#### 3.2.1. Error Recovery for Assembling Objects

In Section 2.2.3, we compensated the error during grasping and transporting by detecting the center of the grasped object. Using the experiment results regarding the final position according to the releasing motion, we can also compensate the positional change during releasing. For example, for the 100 µm microbeads, objects released by dynamic releasing are placed to the right side of the desired position. To compensate the positional change by the releasing motion, the stage moves to the right by approximately 10 µm.

The left image in Figure 11 shows the misplaced object. To arrange several objects precisely, we compensate the positional error from the grasping and transporting process based on the positions of the prearranged objects. Figure 11a describes the error recovery method for compensating the position error based on the prearranged objects. By calculating the distance between the center of the placed object and the grasped object, we compensate the position error. Finally, the target object was moved to the right automatically. The right picture in Figure 11a shows the well-arranged objects based on the prearranged object. We extend this approach to arrange objects as 2D arrays. Figure 11b displays a 2 × 2 array using four objects. To place the fourth object, two arranged objects—the object placed above for adjusting the X-axis and the object placed on the left for adjusting the Y-axis—were employed as a reference.

To make a 3D structure, the objects should be arranged at close distances. Our microhand is a suitable tool for manipulating at close distances because the fingertip is sharp. From Figure 11, we can verify the close distance between objects. The distance between two objects is approximately 20 µm and the objects are well arranged in 122 µm of square.

#### 3.2.2. Automated Assembly of 2D and 3D Structure

To construct 2D and 3D assemblies, 85−120 µm microbeads are utilized. To manipulate the microbeads, 10× lens is utilized, which makes the visible workspace 512 µm × 512 µm. Unlike the manipulation of an object, the microhand lifted the grasped object to 130 µm above the substrate by changing the motorized stage in order to prevent collisions with the prearranged objects. In addition, to manipulate different sizes of objects, we adjusted the Z-position according to the object size. To grasp the object firmly, the end effector should grasp the middle of the object. Since our target object is a sphere, the distance between the middle of the object and the substrate is half the diameter of the object. Thus, we can adjust the grasping point, the middle of the object, based on the object size.

Figure 12a shows the manipulated objects for making assembled “O” and “U” patterns (shown in Appendix A). In addition, we succeed in assembling a 3 × 3 array for a 2D structure and one pyramid composed of five objects for a 3D structure, as shown in Figure 12b,c and Appendix A. The total transporting distance of nine objects for 2D structures is approximately 5.6 mm during 34 s. To create a 3D structure constructed from five objects, the distance of transportation was approximately 1.54 mm, and the manipulation time from the grasping task of the first object to the end of the releasing task of the last object was about 13 s. The average duration of the nine objects and the five objects was 3.7 s/sphere while transporting at 0.62 mm/sphere and 2.6 s/sphere while transferring at 0.3 mm/sphere.

#### 3.2.3. Automated Manipulation Using Spheroids

The assembly method using microbeads can be extended to tissue engineering employing living cells. Control of biological cells is more difficult than that of microbeads because of different physical characteristics. Living cells are stickier than microbeads, which makes the releasing task harder. In addition, the cells are sensitive and can be easily damaged during manipulation. In this study, we utilize NIH3T3 spheroids constructed from 3D cultural plates (Elplasia MPc 500 6, Kuraray, Osaka, Japan) to assemble a pattern as a 1D string.

Figure 13 shows the result of the manipulation of three objects using microbeads and spheroids. The total transporting distance of the two types of objects is approximately 1 mm. The manipulation times of the microbeads and spheroids are 9.2 s and 11.2 s, respectively. The manipulation time of the spheroids is longer than that of microbeads because it takes more time to release the spheroids. To release the spheroids without fail, we increase the performance time of the releasing motion from 50 ms to 100 ms since the amount of the adhesion force between the spheroid and the end effector is different according to the squeezing amount of the spheroid.

Different from the microbeads, spheroids are spontaneously formed from NIH3T3 cells by self-assembly. Thus, the surface of a spheroid can be easily torn by the sharpened glass needle, which causes the failure of the manipulation of the spheroid. In our case, during a transporting task after release of the spheroid, the released spheroid and the microhand sometimes move together even though the spheroid is already separated. The reason is that the surface of the spheroid is torn, and the torn spheroid is attached to the tip of end effector even though the main part of the spheroid is detached. To solve this problem, a potential solution is to change the tip of end effector so that it does not damage the surface of the object during manipulation like [38]. Another solution is to coat the spheroid with hydrogel [39] so that it is not torn by the sharp tip of the end effector.

## 4. Conclusions

This paper presented a high speed and precise assembly method by using dexterous and fast microhand motion. To manipulate objects quickly and precisely, an automated manipulation method was proposed. To achieve this technique, stable grasping and a high-speed transporting system were realized. By using dexterous microhand motions and an automated stage, a system for firm grasping and a large transporting distance (13 mm) at high speed (1 mm/s) can be achieved. In addition, an automatic releasing and error recovery system were addressed for successful manipulation and precise positioning. As a result, 100 µm microbeads were manipulated within 800 ms. To construct a 3D structure, the manipulation of several objects that can transport objects across a large distance was proposed. Finally, we assembled five objects for 3D structures by transporting 1.54 mm within 13 s. This manipulation speed was faster than the highest speed previously reported. In the future, we will apply a path-planning method to speed up the assembly system.

## Figures and Tables

**Figure 1 micromachines-11-00534-f001:**
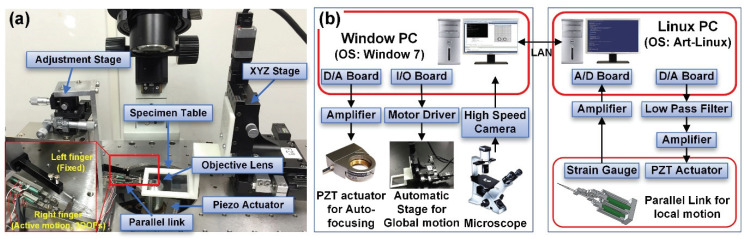
System of microhand. (**a**) Overall system of microhand; (**b**) System architecture.

**Figure 2 micromachines-11-00534-f002:**
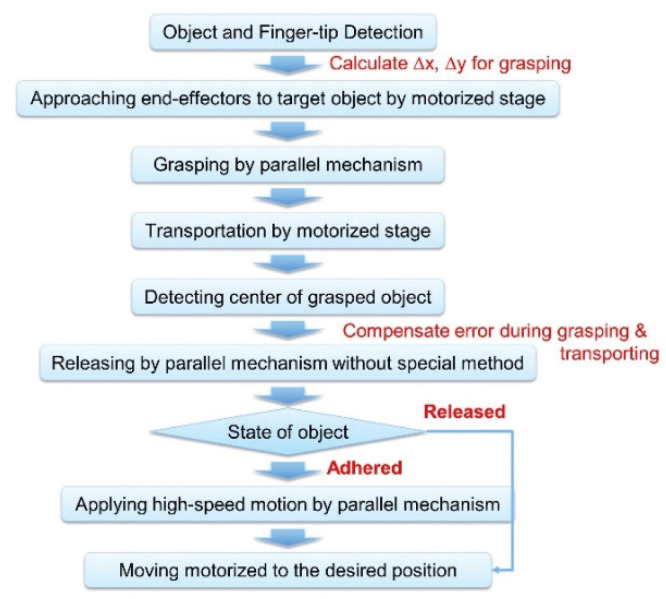
Manipulation process of an object.

**Figure 3 micromachines-11-00534-f003:**
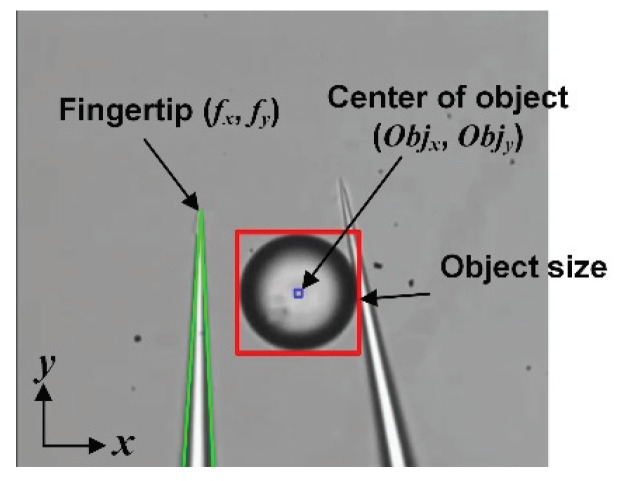
Position detection of the left fingertip and the center of an object.

**Figure 4 micromachines-11-00534-f004:**
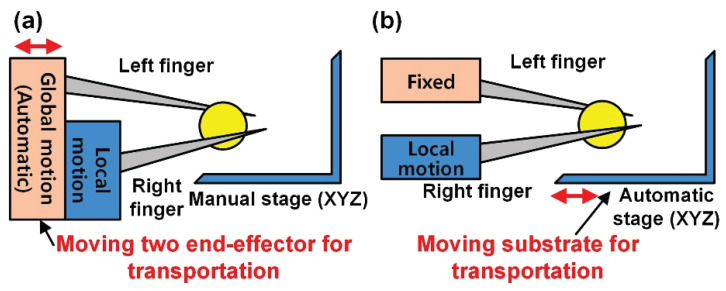
Transportation system. (**a**) Moving two end effectors; (**b**) Moving substrate.

**Figure 5 micromachines-11-00534-f005:**
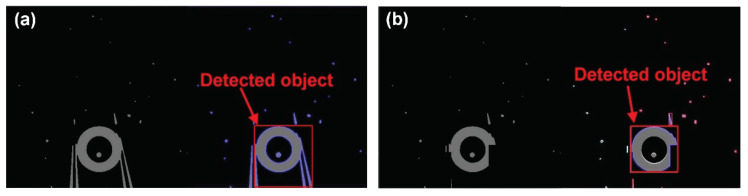
Detecting the grasped objects. (**a**) Before removing two fingers; (**b**) After removing two fingers.

**Figure 6 micromachines-11-00534-f006:**
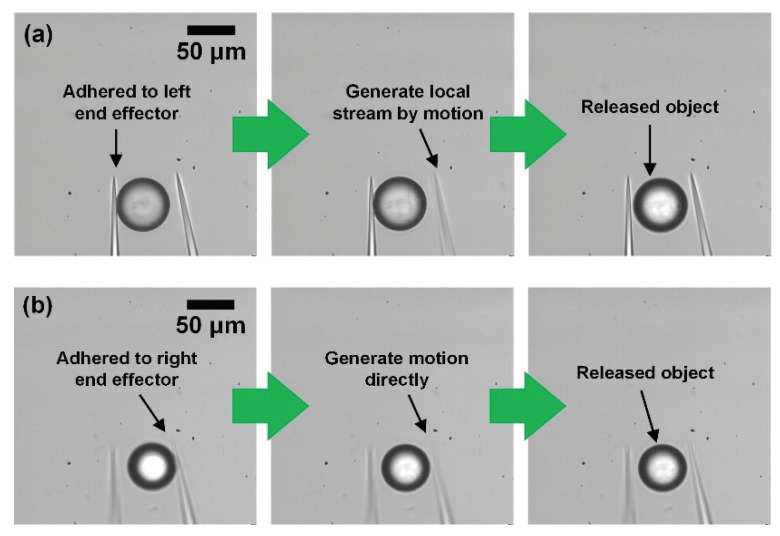
Automatic releasing (**a**) When the object is adhered to left; (**b**) When the object is adhered to right.

**Figure 7 micromachines-11-00534-f007:**
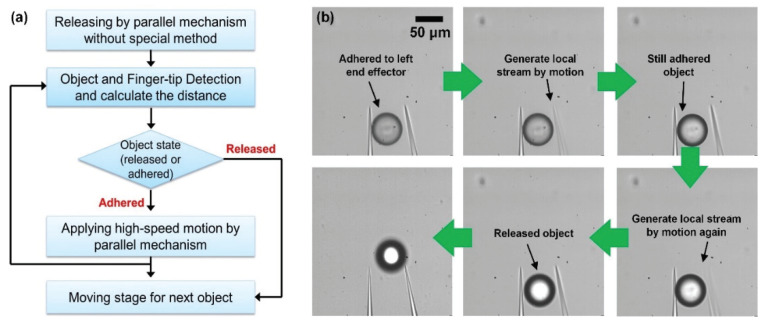
Releasing process for improving success rate. (**a**) Flowchart; (**b**) Image sequence of releasing process.

**Figure 8 micromachines-11-00534-f008:**
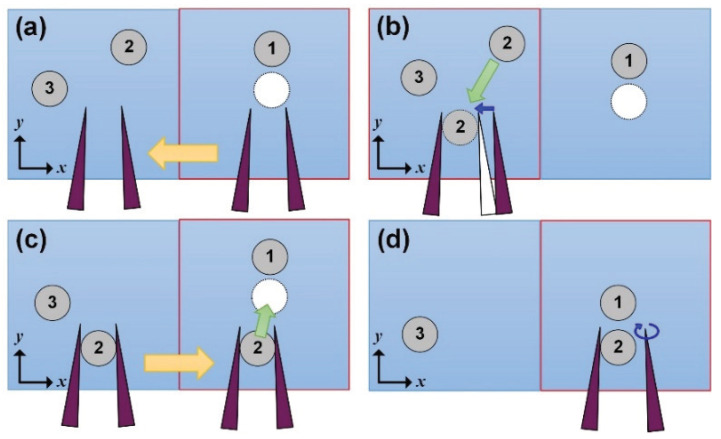
Manipulation of several objects outside of the visible space. (**a**) Moving the stage to find the second object; (**b**) Searching and grasping the second object; (**c**) Transporting the second object to the desired position; (**d**) Releasing the second object.

**Figure 9 micromachines-11-00534-f009:**
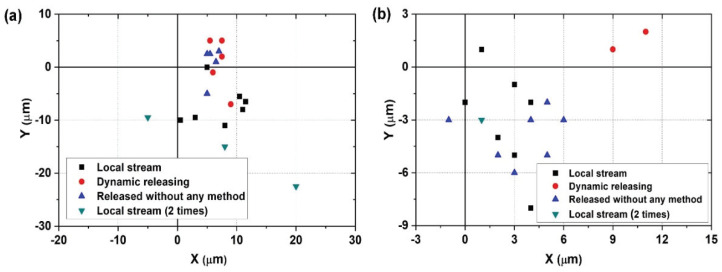
Result of the automated manipulation of an object; (**a**) Placing position of 100 µm microbeads; (**b**) Placing position of 55 µm microbeads.

**Figure 10 micromachines-11-00534-f010:**
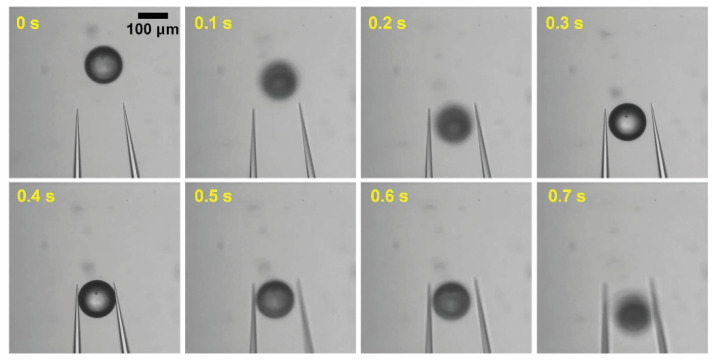
Time-lapse images of the automated manipulation of an object.

**Figure 11 micromachines-11-00534-f011:**
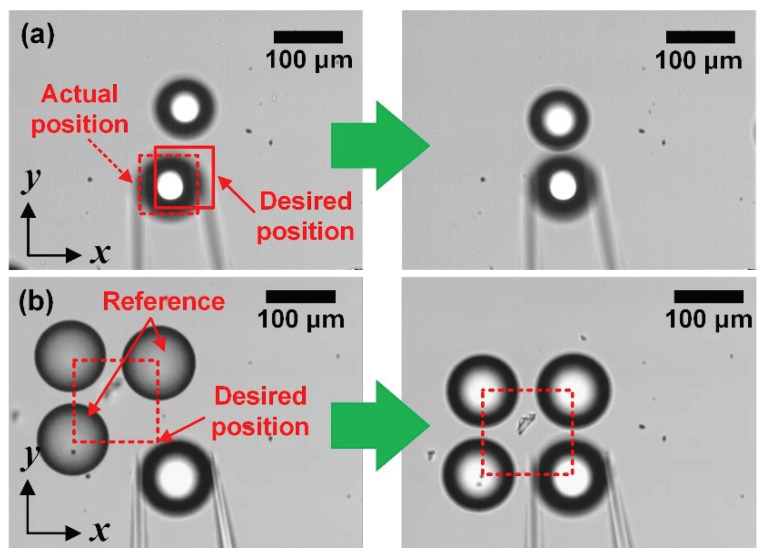
Error recovery to compensate the position error based on a prearranged object. (**a**) Manipulating the second object by adjusting X-axis; (**b**) Manipulating the fourth object by adjusting X and Y-axis.

**Figure 12 micromachines-11-00534-f012:**
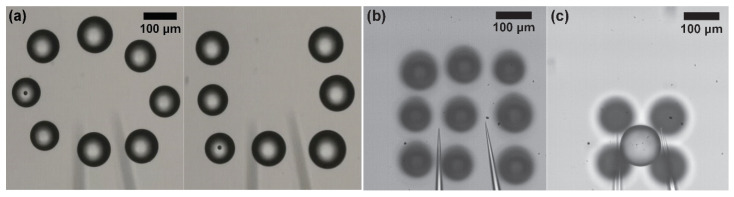
Results of the manipulation. (**a**) Several objects for making special characters (“O” and “U”); (**b**) Nine objects for making 2D structure; (**c**) Five objects for making 3D structure.

**Figure 13 micromachines-11-00534-f013:**
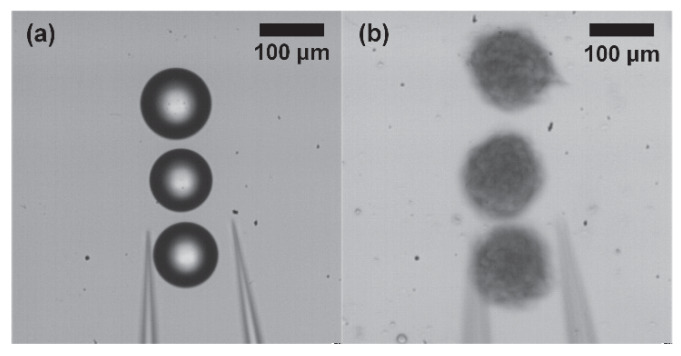
Results of the manipulation of three objects. (**a**) 85−120 µm microbeads, (**b**) 100−120 µm spheroids.

**Table 1 micromachines-11-00534-t001:** Automated high speed pick-and-place method in air and liquid.

Parameter	Air	Liquid		
	Xie et al. 2009	Zhang et al. 2010	Lu et al. 2010	Avci et al. 2015
Transfer speed	NA	82 µm/s	0.2 mm/s	2.3 mm/s
Transfer distance	9 µm/sphere	77 µm/sphere	1 mm/sphere	60 µm/sphere
Total speed	48 s/sphere	6 s/sphere	15 s/sphere	1 s/sphere
Success rate	100%	100%	95.13%	84%
Structure	3D pyramid	2D patterns	2D (microwell)	1D string

**Table 2 micromachines-11-00534-t002:** Average required time for manipulation of 55 µm and 100 µm microbeads.

Manipulation Process	55 µm Microbeads	100 µm Microbeads
Detecting and approaching	125 ms	170 ms
Grasping	100 ms	100 ms
Transporting	105 ms	105 ms
Releasing	382.5 ms	330 ms
**Total**	**712.5 ms**	**705 ms**

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
