# Peer review of "High-Speed Manipulation of Microobjects Using an Automated Two-Fingered Microhand for 3D Microassembly"

_micromachines, 2020, doi:10.3390/mi11050534_

Round 1
Reviewer 1 Report
The paper by KiM and et al titled "High-speed Manipulation of Microobjects by using 3 Automated Two-fingered Microhand for 3D 4 Microassembly " is well written with nice results. Before it can be accepted for publication, I really have several concerns.
First of all, I think the author made a mistake about the name of their university, "Beijin" should be " Beijing". Also, please check your main text carefully. The writing of this paper need more efforts.
Secondly, the paper is lack of scientific story. the organization is messy with 17 figures. It looks like a application note for company product. I suggest that the author can combine them to several figures to show the outline of your paper clearly.
Finally, Maybe more quantitatively analysis is needed . For example, does different size beads will have different speed?Is there any limit on the beads size?
Author Response
We are very much thankful to the reviewers for their deep and thorough review. We have revised our present research paper in the light of their useful suggestions and comments. We hope our revision has improved the paper to a level of their satisfaction. Number wise answers to their specific comments are as follows.
Comment 1: First of all, I think the author made a mistake about the name of their university, "Beijin" should be "Beijing". Also, please check your main text carefully. The writing of this paper need more efforts.
Response: We changed the mistake from Beijin to Beijing. In addition, we checked our main text carefully.
Comment 2: Secondly, the paper is lack of scientific story. the organization is messy with 17 figures. It looks like a application note for company product. I suggest that the author can combine them to several figures to show the outline of your paper clearly.
Response: We combined several figures to show the outline of our paper clearly (Figure 1 and 2, Figure 7 and 8, Figure 9 and 10 and, Figure 13 and 14). Finally, we reduced the figure from 17 to 13 (including one additional figure).
2.1 System for micromanipulation, Figure 1 (combine Figure 1 and 2)
2.2.4. Releasing and Precise Positioning of Objects, Figure 6 (combine Figure 7 and 8) and Figure 7 (combine Figure 9 and 10)
3.2.1. Error Recovery for Assembling Objects, Figure 11 (combine Figure 13 and 14)
Comment 3: Finally, Maybe more quantitatively analysis is needed. For example, does different size beads will have different speed? Is there any limit on the beads size?
Response: We compared the manipulation speed of two microbeads, 55 µm and 100 µm microbeads, in section 3.1. For this, we also added one figure and changed the table 2. Our manipulation method can manipulate 10-120 µm microbeads (line 125). However, in this research, we focused on 80-120 µm microbeads for our target, NIH3T3 spheroid. Thus, we also add explanation of the target object in line 24 and 52.
3.1. Automated Manipulation of an Object, line 320 – 355
Abstract, line 24
Finally, we succeed in assembling 80 - 120 μm of microbeads and 23 spheroids integrated by NIH3T3 cells.
Introduction, line 51
The target of this research is 100 µm spheroids composed by NIH3T3 cells.
Introduction, line 112
5) manipulation of NIH3T3 spheroids for tissue engineering.
2.1. System for micromanipulation, line 125
To manipulate micro objects having different diameters, a large distance between the two end-effectors should be realized. The proposed microhand is able to manipulate objects from 10 µm to 120 µm of diameter.

Reviewer 2 Report
The authors present an automated two-fingered microhand for high-speed manipulation of microobjects with high accuracy. This was an improvement of their earlier work. They use a vision based automatic manipulation for assembly of 2D and 3D structure at high speed for tissue engineering applications.
For the benefit of the readers, the reviewer suggested the following important revisions:
- Discuss what actuation technologies (electrostatic, piezo) are currently employed by adding few references.
- Include data in the form of time-lapse figures that highlight the high speed actuation. In addition, have a short discussion.
- Fig. 6 needs some improvement
Author Response
We are very much thankful to the reviewers for their deep and thorough review. We have revised our present research paper in the light of their useful suggestions and comments. We hope our revision has improved the paper to a level of their satisfaction. Number wise answers to their specific comments are as follows.
Comment 1: Discuss what actuation technologies (electrostatic, piezo) are currently employed by adding few references.
Response: We added several reference with regard to micromanipulation applying a robotics approaches.
Introduction – Second paragraph, line 37-46
Manipulation applying a robotics approaches have been discussed. It can be divided into two categories: non-contact and contact manipulation. Non-contact manipulation utilizes field forces such as acoustic wave [11-13], dielectrophoresis [14], optical tweezers [15], and local flow [16-17]. Contactless manipulation is able to handle multiple objects, at the same time, and is not affected by adhesion forces. However, it cannot perform physical operation tasks and difficult to assemble objects as a 3D structure. On the other hand, contact manipulation directly contacts with target objects by employing several actuation, for example, a vacuum tool [18], voltage control [19], piezoelectric actuator [7, 20], and plunging mechanism [21]. Although this method can be applicable for building 3D microsystems and devices [22], the main difficulties are releasing of micro-objects due to strong adhesion forces.
Comment 2: Include data in the form of time-lapse figures that highlight the high speed actuation. In addition, have a short discussion.
Reponse: We added time-lapse images of the automated manipulation of an object in Figure 10.
3.1 Automated Manipulation of an Object, Figure 10
Comment 3: Fig. 6 needs some improvement
Response: We changed the Figure 6. We add a figure that include the detected object before removing two fingers to compare two cases, before and after removing two fingers.
Section 2.2.3 compensating Error during Grasping and Transporting, Figure 5

Round 2
Reviewer 2 Report
Thanks for making the suggested changes.